# The Pathway-Selective Dependence of Nitric Oxide for Long-Term Potentiation in the Anterior Cingulate Cortex of Adult Mice

**DOI:** 10.3390/biomedicines12051072

**Published:** 2024-05-12

**Authors:** Qi-Yu Chen, Jinjin Wan, Yujie Ma, Min Zhuo

**Affiliations:** 1CAS Key Laboratory of Brain Connectome and Manipulation, Interdisciplinary Center for Brain Information, Chinese Academy of Sciences Shenzhen Institute of Advanced Technology, Shenzhen 518055, China; 2Zhuomin International Institute for Brain Research, Qingdao 266000, China; 3Oujiang Laboratory, Wenzhou Medical University, Wenzhou 325027, China; 4Department of Physiology, Faculty of Medicine, University of Toronto, Medical Science Building, Room #3342, 1 King’s College Circle, Toronto, ON M5S 1A8, Canada

**Keywords:** nitric oxide, anterior cingulate cortex, synaptic potentiation, long-term potentiation

## Abstract

Nitric oxide (NO) is a key diffusible messenger in the mammalian brain. It has been proposed that NO may diffuse in retrograde into presynaptic terminals, contributing to the induction of hippocampal long-term potentiation (LTP). Here, we present novel evidence that NO is selectively required for the synaptic potentiation of the interhemispheric projection in the anterior cingulate cortex (ACC). Unilateral low-frequency stimulation (LFS) induced a short-term synaptic potentiation on the contralateral ACC through the corpus callosum (CC). The use of the antagonists of the NMDA receptor (NMDAR), or the inhibitor of the L-type voltage-dependent Ca^2+^ channels (L-VDCCs), blocked the induction of this ACC-ACC potentiation. In addition, the inhibitor of NO synthase, or inhibitors for its downstream signaling pathway, also blocked this ACC-ACC potentiation. However, the application of the NOS inhibitor blocked neither the local electric stimulation-induced LTP nor the stimulation-induced recruitment of silent responses. Our results present strong evidence for the pathway-selective roles of NO in the LTP of the ACC.

## 1. Introduction

Synaptic long-term potentiation (LTP), a typical synaptic model, has been widely studied within and between various cortical and subcortical regions, including the hippocampus and cerebral cortices [1,2,3,4]. There are three different forms of LTP at least that have been reported in the hippocampus and cortex regarding basic mechanisms for the induction and expression of LTP. In the anterior cingulate cortex (ACC), two major forms of LTP have been reported in adult cortical synapses: NMDA receptor (NMDAR)-mediated postsynaptic LTP (post-LTP) and kainate receptor (KAR)-mediated presynaptic LTP (pre-LTP) [5]. The induction of post-LTP requires the NMDARs and the L-type voltage-dependent Ca^2+^ channels (L-VDCCs). The induction of pre-LTP in the ACC requires KARs but not NMDARs. In the hippocampus, in addition to the post- and pre-LTP, there is another form of LTP that requires retrograde messengers (e.g., nitric oxide (NO), carbon oxide (CO), and arachidonic acid) [6,7,8,9,10]. By pairing the electric stimulation on the presynaptic fibers with NO/CO application, LTP was recorded in the CA1. This LTP was further found to be independent of the NMDARs and L-VDCCs. However, it is still unknown whether this retrograde messenger-mediating LTP exists in the ACC.

Nitric oxide (NO) is a ubiquitous signaling molecule in the brain and other organs of the body. As one of the candidates of retrograde messengers, NO plays a significant role in synaptic potentiation [9,10,11,12]. NO is synthesized from L-arginine by nitric oxide synthases (NOSs). Early studies have found that the application of inhibitors of NOS extracellularly or intracellularly in the hippocampus blocks the induction of LTP [7]. It has been proposed that NO may be produced at postsynaptic sites by the activation of postsynaptic NMDARs and then diffuses to presynaptic terminals in which it activates the cyclic guanosine monophosphate (cGMP)-dependent signaling pathway and further enhances the release of glutamate during LTP [9,10,11,12]. 

Cumulative evidence has found that the anterior cingulate cortex (ACC) plays important roles in the coding of pain and unpleasantness in both human and animal models [1,4,13]. The projectome of the ACC has been well established [14,15]. A recent study has proved that direct inter-hemispheric projection between the left and right ACC plays a critical role in chronic pain [15]. It has been proved that NOS is expressed in the ACC [16,17,18]. However, the roles of NO in synaptic potentiation in the ACC have not been elucidated. Therefore, in this study, we will investigate the role of NO in the synaptic potentiation in the pyramidal neurons in the ACC, especially in ACC-ACC potentiation.

## 2. Materials and Methods

### 2.1. Animals 

Adult male and female C57BL/6 mice (7–9 weeks old) were used. All animals were housed under a 12 h light–dark cycle with food and water provided ad libitum. Experiments were conducted under the protocol approved by the Animal Care and Use Committees at the Oujiang Laboratory (OJLAB22121612).

### 2.2. Drugs

D-(-)-2-Amino-5-phosphonopentanoic acid (AP-5), Nomega-Nitro-L-arginine methyl ester hydrochloride (L-NAME), (S)-1-(2-Amino-2-carboxyethyl)-3-(2 -carboxy-thiophene-3-yl-methyl)-5-methylpyrimidine-2,4-dione (UBP 310), picrotoxin, and Ro 25-6981 were purchased from HelloBio (Princeton, NJ, USA). Nimodipine, ODQ, BAPTA, Rp-8-pCPT-cGMP, and picrotoxin were bought from Sigma-Aldrich (St Louis, MO, USA). Drugs were prepared as stock solutions for frozen aliquots at −20 °C. All drugs were diluted from the stock solution to the final concentration in the ACSF before being applied to brain slices.

### 2.3. Brain Slices Preparation

Coronal brain slices (300 μm) of ACC were prepared using standard methods [19,20]. Briefly, mice were deeply anesthetized with 5% isoflurane and sacrificed by decapitation. The whole brain was removed quickly from the skull and submerged in the oxygenated (95% O_2_ and 5% CO_2_) ice-cold artificial cerebrospinal fluid (ACSF) containing (in mM) 124 NaCl, 2.5 KCl, 2 MgSO_4_, 1 NaH_2_PO_4_, 2 CaCl_2_, 25 NaHCO_3_, and 10 D-glucose. Coronal slices were prepared by the vibratome (VT1200S Vibratome, Leica, Germany). Slices were then transferred and incubated in a submerged recovery chamber at room temperature for one hour. The ACSF was continuously balanced with a mixture of 95% O_2_ and 5% CO_2_.

### 2.4. MED64 Recording

After incubation, slices were transferred to the recording chamber and perfused with ACSF at 28–30 °C. The slices were positioned on the MED64 (MED64, Alpha-Med Sciences, Osaka, Japan) P515A probe in such a way that the whole array of the electrodes could cover the different layers of the ACC. One column of the channels located in layer II/II or layer V of the ACC, from which the best synaptic responses could be induced in the surrounding recording channels, was then chosen as the stimulation site. Slices were kept in the recording chamber for at least 1 h before the start of the experiments. Bipolar constant current pulse stimulation (1–10 mA, 0.2 ms) was applied to the stimulation channel and the intensity was adjusted so that a half-maximal field excitatory postsynaptic potential (fEPSP) was elicited in the channels closest to the stimulation site. The channels with fEPSP were considered active channels and their fEPSP responses were sampled every 1 min and averaged every 4 min. The parameter of ‘slope’ indicated the average slope of each fEPSP recorded by activated channels. Stable baseline responses were first recorded within 1 h. Then, a TBS (five trains of bursts with four pulses at 100 Hz at 200 ms intervals; repeated five times at intervals of 10 s) was applied to the same stimulation channel to induce LTP. 

### 2.5. Whole-Cell Patch-Clamp Recording

Whole-cell recordings were performed in a recording chamber on the stage of an Olympus BX51 microscope with infrared differential interference contrast (DIC) optics for visualization. EPSCs were recorded from layer II/III neurons with an Axon 200B amplifier (Molecular Devices, San Jose, CA, USA), and the stimulations were evoked in layer V of the ACC by a bipolar tungsten-stimulating electrode. The recording pipettes (3–5 MΩ) were filled with solution containing (in mM) 145 K-gluconate, 5 NaCl, 1 MgCl_2_, 0.2 EGTA, 10 HEPES, 2 Mg-ATP, and 0.1 Na_3_-GTP, which adjusted to pH 7.3 with KOH and had osmolality of 300 mOsmol. The amplitudes of evoked EPSCs were adjusted to between 100 and 150 pA to obtain a baseline. Picrotoxin (PTX, 100 μm) was always present to block the GABA_A_ receptor-mediated inhibitory synaptic currents in all experiments. Access resistance was 15–30 MΩ and monitored throughout the experiment. Data were collected only when access resistance changed <15% during all experiments. Data were filtered at 1 kHz and digitized at 10 kHz.

### 2.6. Statistical Analysis

OriginPro 8.0 (Originlab Corporation, Northampton, MA, USA) was used for plotting figures, and SPSS version 22.0 (SAS Institute Inc, Cary, NC, USA) software was used to analyze the results. The paired *t*-tests or one-way ANOVA were conducted as appropriate. All data were presented as the mean ± standard error of the mean (SEM). In all cases, *p* < 0.05 was considered statistically significant. 

## 3. Results

### 3.1. Low-Frequency Stimulation Induces Synaptic Potentiation in the ACC

By using whole-cell patch-recording, we first applied low-frequency stimulation (LFS, 1 Hz for 60 s) in the left ACC and recorded the spontaneous excitatory postsynaptic current (sEPSC) of the right ACC neurons to test whether contralateral LFS could induce synaptic potentiation in the ipsilateral ACC neurons. The interpretation of sEPSC amplitude is complicated by the fact that these events represent a mixture of miniature and evoked (by spontaneous cell firing) synaptic currents. Early studies have proved that the frequency of the sEPSC represents the presynaptic release of the neurotransmitters [21]. We performed whole-cell patch-clamp recording here and patched the neurons at −70 mV; therefore, the increased frequency of the sEPSC indicates the enhanced probability of spontaneous glutamate release. Interestingly, contralateral LFS induced an increase in the frequency of the sEPSC (Figure 1a). Figure 1b demonstrates the effects of contralateral LFS on the cumulative distribution of the inter-event interval and amplitude of the sEPSCs. Contralateral LFS increased the proportion of sEPSCs having a shorter inter-event interval (*p* < 0.05) but did not change the amplitude (*p* < 0.05) when compared with that before the stimulation (same neuron as in Figure 1a). We found that contralateral LFS induced the synaptic potentiation of the frequency (Figure 1c,e), but not the amplitude of the sEPSCs (Figure 1d,g). This indicates that presynaptic glutamate release was enhanced by the LFS. Next, to observe whether a longer duration of the LFS or higher stimulation frequency had a similar effect, we used 1 Hz for 180 s, 10 Hz for 10 s, 100 Hz for 1 s, and theta burst stimulation (TBS) on the contralateral ACC. As shown in Figure 1f,h, only 1 Hz for 60 s stimulation induced potentiation significantly in the ACC, and none of the stimulation protocols above changed the amplitude. 

### 3.2. The LFS-Induced Synaptic Potentiation in the ACC Is Induced through the Corpus Callosum

A recent work has proved that interhemispheric projection through the corpus callosum (CC) in the ACC exists [15]. To investigate whether the contralateral LFS induces synaptic potentiation through the CC, we cut the midline of the coronal brain slices (Figure 2a). The brain slices cut through the midline did not show potentiation (Figure 2b), but the slices cut under the CC showed potentiation (Figure 2b,c). Neither group showed changes in the amplitude in the sEPSCs (Figure 2d,e). We measured the sEPSC for 30 s in the cutting under the CC group; the average increases in EPSC frequency and amplitude were 150.3 ± 17.6% and 100.5 ± 4.3% (n = 8), respectively (Figure 2f,g). These data further confirm that ACC-ACC potentiation is mediated by the projections in the CC. 

### 3.3. The Synaptic Potentiation of ACC-ACC Is Dependent on the NMDA Receptor

Previous studies found that the induction of LTP in the ACC requires NMDA receptors (NMDARs) and the L-type voltage-dependent Ca^2+^ channels (L-VDCCs) [1,22]. To investigate whether the synaptic potentiation of ACC-ACC requires the NMDARs, we applied an antagonist of NMDARs (AP-5, 50 μM) in the bath solution. The potentiation was blocked by the presence of AP-5 (Figure 3a), indicating that the synaptic potentiation of ACC-ACC is NMDAR-dependent. To further address the possible involvement of L-VDCC, we used an inhibitor of the L-VDCC (Nimodipine, 10 μM) and we found that Nimodipine also blocked the potentiation, indicating that L-VDCCs are important for ACC-ACC potentiation (Figure 3b). We also focused on kainate receptors (KARs), which play key roles in the presynaptic LTP in the ACC [5]. We applied a specific antagonist of the GluK1 receptor, UBP310 (10 μM), to the adult ACC slices, and we found that UBP310 did not affect the synaptic potentiation of ACC-ACC connection (Figure 3c). Furthermore, we investigated whether GluN2B subunit-containing NMDAR is required for the synaptic potentiation of ACC-ACC. We applied an antagonist of GluN2B-containing NMDAR, Ro 25-6981 (0.3 μM). Ro 25-6981 completely blocked ACC-ACC potentiation (Figure 3d). There were no significant changes among AP-5, nimodipine, and Ro 25-6981 (Figure 3e,f). These results suggest that the synaptic potentiation of ACC-ACC requires NMDAR, L-VDCC, and GluN2B.

### 3.4. Nitric Oxide Is Critical for the Synaptic Potentiation of the ACC-ACC Connection

To test whether NO participates in ACC-ACC potentiation, we applied an inhibitor of NOS, L-NAME (100 μM), to the bath solution. The ACC-ACC potentiation did not show in the presence of L-NAME (Figure 4a,c), indicating that NO is involved in the synaptic potentiation of the ACC-ACC connection. It is known that a Ca^2+^ signaling pathway is required for NO synthesis, and thus we inhibited postsynaptic Ca^2+^ signaling by applying an internal Ca^2+^ chelator (BAPTA, 20 mM) in the recording pipette. Interestingly, the BAPTA failed to block the potentiation (Figure 4b,c), indicating that the NO may not be synthesized by the recording postsynaptic neurons; instead, neurons around the recording neuron produced NO.

To assess the impact of NO on spontaneous excitatory synaptic transmission, we applied L-NAME (100 μM) to the bath solution. The application of L-NAME resulted in neither the frequency nor the amplitude of sEPSCs being affected (Figure 4e,f). This result suggests that NO does not affect basal transmission in the ACC.

### 3.5. NO Participates in the Synaptic Potentiation between ACC and ACC via the sGC Signaling Pathway

To further confirm the role of NO in ACC-ACC potentiation, we applied inhibitors downstream of the NO. The bath application of the inhibitor of the soluble guanylate cyclase (sGC), ODQ (10 μM), blocked potentiation in the ACC (Figure 5a,c). Next, we also applied an inhibitor for the protein kinase G (PKG), Rp-8-pCPT-cGMP (1 μM), to examine whether PKG activation is required for potentiation in the ACC. We found that Rp-8-pCPT-cGMP blocked ACC-ACC potentiation on the frequency of the sEPSCs (Figure 5b,c). These results suggest that the cGMP/PKG retrograde signaling pathway is critical for ACC-ACC potentiation.

### 3.6. NO Is Not Required for homo-LTP within the ACC

To further prove that NO-dependent synaptic potentiation is selectively dependent on the ACC-ACC connection, we also investigated the effect of NOS inhibition on the LTP induced by the local stimulation in the ACC. By using the multi-electrode recording system (MED64, Figure 6a), we found that TBS successfully induced homosynaptic LTP in four out of six activated channels in one ACC slice in the presence of L-NAME (Figure 6b–e). The slope of the TBS-induced LTP in the ACC reached a comparable level to the control group (Figure 6f–i). These data indicate that the homo-LTP in the ACC does not require NO. 

Previous studies found that TBS induced the recruitment of silent channels by using a MED64 recording system [23,24]. We further analyzed the effect of NO on the recruitment of silent channels. Recruited channels were found on the ACC slices after the TBS in the presence of L-NAME (Figure 7a,b). The number and the average amplitude of the recruited channels had no significant difference compared with those of the saline group (Figure 7c,d). These data indicate that the recruitment of silent responses does not require NO. 

## 4. Discussion

In this study, we discovered a short-lasting potentiation in the ACC that is dependent on NO. This potentiation is selective to the ACC-ACC pathway. By using whole-cell patch-clamp techniques, we further found that this ACC-ACC potentiation is dependent on the GluN2B-containing NMDAR, L-VDCC, and NO. As a comparison, local stimuli-induced homosynaptic LTP in the ACC does not require NO. This is the first time that it has been demonstrated that ACC-ACC potentiation is selectively dependent on NO.

### 4.1. LFS-Induced Synaptic Potentiation

In our study, we used different stimulation protocols on the contralateral ACC. Short-term potentiation was induced by 1 Hz 60 s. A longer duration of LFS or TBS did not induce a similar effect. The LFS can induce potentiation in the amygdala, hippocampus, and cortex [5,25,26]. However, most LFS-induced potentiation is independent of the NMDAR due to stimulation at a frequency of 1 Hz being well beyond the time window for eliciting NMDAR-dependent potentiation [27,28,29]. Nevertheless, some forms of NMDAR-dependent potentiation showed wider temporal windows for the integration of synaptic inputs to trigger synaptic enhancement [27,28,29,30]. The synaptic potentiation in CA1 elicited by LFS on the commissural and medial septal afferents requires an NMDAR [30]. The ACC-ACC potentiation recorded in our study also showed the NMDA receptor- and L-VDCC-dependence. This evidence further indicates that different types of LFS-induced potentiation exist in the central nervous system, while the details of intracellular mechanisms are still unknown.

### 4.2. ACC-ACC Potentiation vs. Pre-LTP in the ACC

A recent study has demonstrated that the callosal projection between the left and right hemispheric ACC forms a positive feedback loop for excitatory transmission [15]. Single postsynaptic responses can be induced by electrical or light stimulation of ACC-ACC projection. These postsynaptic responses are mediated by pure AMPA receptors. In our study, we first placed the stimulating electrode on the contralateral ACC. By cutting the coronal slices on the CC or under the CC, we further confirmed that ACC-ACC potentiation is mediated by CC connection. This is in accordance with the morphological evidence that the ACC-ACC connection was passed through the CC. 

In the ACC, LFS induces presynaptic LTP, indicated by increased evoked EPSC and decreased paired-pulse ratio [5]. The increased frequency of sEPSC and unaltered amplitude in our study further confirmed that the LFS applied on the contralateral ACC caused presynaptic facilitation. However, the presynaptic LTP in the ACC does not require NMDAR. The following pharmacological studies in our study further demonstrated that the ACC-ACC potentiation was dependent on the NMDA receptors and L-VDCC. This is consistent with the theory for the postsynaptic LTP in the ACC [31]. The independence of GluK1 in this ACC-ACC potentiation shows another, different feature from the presynaptic LTP in the ACC [5]. Although the ACC-ACC potentiation shows presynaptic enhancement, the induction protocols are different. 

### 4.3. NO Is Required for the LFS-Induced ACC-ACC Potentiation but Not for the TBS-Induced homo-LTP in the ACC

Previous studies in the hippocampus have reported that the LTP is dependent on NO, which is released from the postsynaptic sites and retrograded and transported to presynaptic terminals. Therefore, this LTP, induced by NO paired with weak tetanic electric stimulation, requires neither postsynaptic NMDA receptors nor the L-VDCC [9]. Interestingly, in the acute application of the inhibitor of NO synthase (L-NAME) in the hippocampus, the effect on the hippocampal LTP can vary depending on the recording sites and dose of the L-NAME [32,33,34,35]. Even in the same neuronal circuits, the suppression of LTP by L-NAME is through temperature [8]. In the cortices, L-NAME has also been reported to affect the induction of the LTP in the somatosensory cortex, the barrel cortex, and the motor cortex [8,36,37]. It is a major finding that NMDAR and neuronal NOS (nNOS) exist in the postsynaptic site [9,10,11,12]. The hypothesis of NO as a retrograde messenger diffusing from the postsynaptic to the presynaptic site is widely known. However, we cannot rule out the possibility that NOS can also be present in presynaptic neurons. In a recent study, by using double patch-clamp techniques, the presynaptic LTP was blocked by either perfusing the L-NAME or injecting MK-801 in the presynaptic neurons in the insular cortex (unpublished paper), which indicates that presynaptic NMDARs as well as their following NOS activation may also exist. 

In our study, under the same recording temperature and using mice of the same age, we found a discrepancy in NO-dependency between the LFS-induced ACC-ACC potentiation and the local TBS-induced LTP in the ACC. We found that a bath application of the selective inhibitor for the NOS blocked the synaptic potentiation mediated by the ACC-ACC connection. As a comparison, we induced the LTP by local TBS stimulation in the ipsilateral ACC of the recorded neurons. Previous studies have reported that ACC neurons receive projections from other cortical areas as well as subcortical nuclei [1,13]. Those corticocortical projections include the ACC-ACC callosal projections. When the TBS is applied to the ACC, a variety of projections to the recorded ACC neuron can be activated. However, the induction of local TBS-induced LTP is not affected by the application of the L-NAME. Therefore, we propose that the ACC-ACC potentiation is selectively dependent on NO. There are two things to note about this difference: (1) NO-dependent synaptic potentiation relies on the selective neuronal pathway in the ACC; (2) LFS-induced potentiation does not share the same mechanism as the TBS-induced LTP. 

It has been proved that NOS can be synthesized in the ACC [16,17,18], and that NO synthesis requires intracellular Ca^2+^. It is curious that the postsynaptic inhibition of Ca^2+^ did not block the ACC-ACC potentiation, but that ACC-ACC potentiation requires a NO signaling pathway. We inferred that the postsynaptic NO was synthesized in other neurons instead of the CC-producing neurons. It has been reported that the application of a NO donor enhanced the spontaneous excitatory responses in hippocampal neurons as well as in the cerebellar cortex [38,39]. We cannot rule out the possibility that the NO donor would yield a change in the sEPSC of the ACC. This requires confirmation in a future study. 

By using the animal model of neuropathic pain, two recent studies about the ACC-ACC connection have reported that the ACC-ACC connection modulates both the ipsilateral and contralateral hindpaw hypersensitivity of the nerve-injured side, as well as the pain-related unpleasantness [15,40]. Several possible positive loops exist in the CNS to enhance or facilitate pain-related information: short-distance positive feedback (ACC-ACC); long-distance positive feedback (ACC-spinal cord); unilateral positive feedback; and bilateral positive feedback. Considering the circuit-like innervation between bilateral ACCs, such unique circuits provide a positive feedback loop for the processing of nociceptive information in the cortex. Thus, pain perception can be reinforced through such short-distance and long-distance projections. We can infer that long-term hyperactivity is related to the enhanced synaptic potentiation in the ACC-ACC connection. The roles of LTP in the ACC in the coding of pain and unpleasantness have been reported in cumulative studies [1,4]. LTP was recorded in the ACC vivo in the contralateral side of the digit amputation in anesthetized rats [41]. In the in vitro ACC slices, LTP was occluded [19]. Future studies will further investigate the role of the NO-related signaling pathway in the ACC of animal models of chronic pain and its related emotional disorders.

## Figures and Tables

**Figure 1 biomedicines-12-01072-f001:**
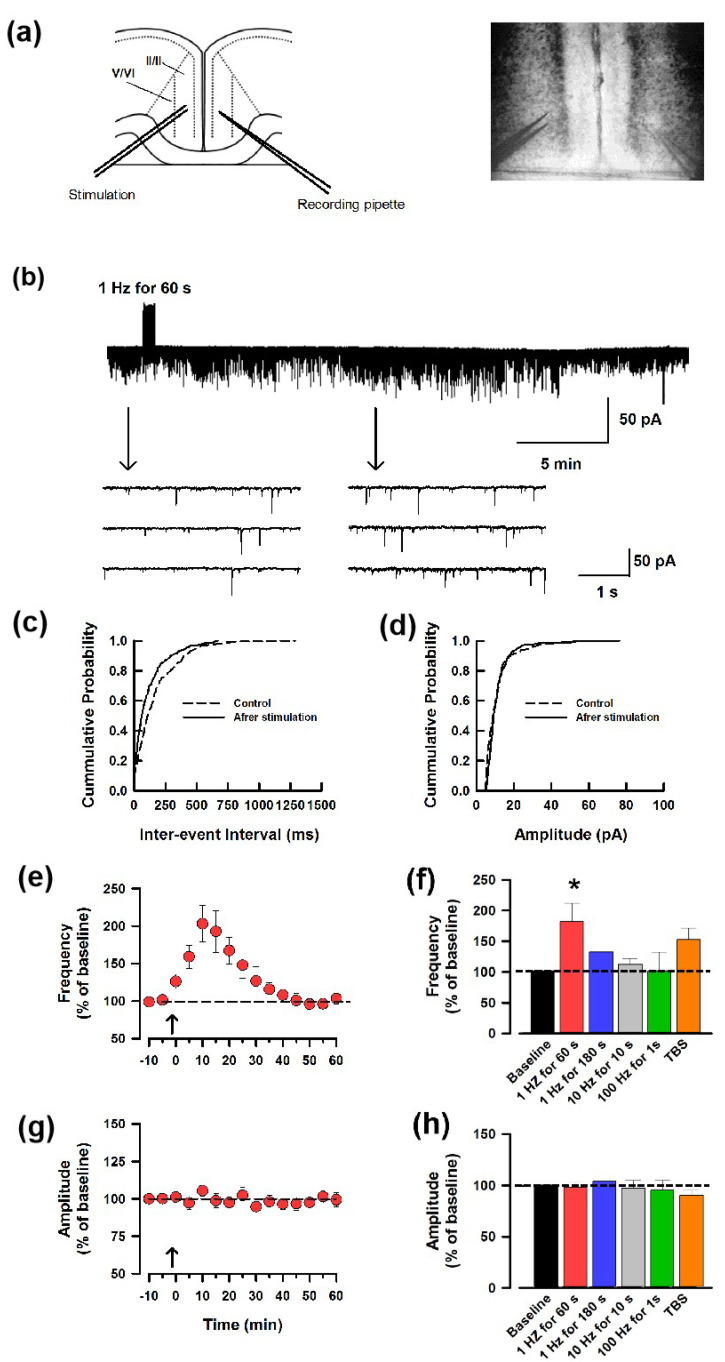
Unilateral LFS facilitates contralateral glutamate release of the ACC. (**a**) Representative recording diagram and photograph showing the placement of the stimulating electrode on one side of the ACC and the recording electrode on the contralateral ACC. (**b**) One sample showed ACC stimulation-induced facilitation of the frequency instead of the amplitude of the sEPSC. (**c**,**d**). Cumulative fraction of inter-event interval (**c**) and amplitude (**d**) of the sEPSCs in the phase of baseline (dash line) and after LFS (black line). (**e**) The average frequency of sEPSC in 1 h. The LFS was applied 10 min after the baseline recording (shown in arrow). (**f**) Summarized results of the frequency of sEPSC from 5 different potentiation-induced protocols (black: baseline; red: 1 Hz for 60 s; blue: 1 Hz for 180 s; grey: 10 Hz for 10 s; green: 100 Hz for 1 s; orange: TBS). * *p* < 0.05, one-way ANOVA. (**g**) Averaged amplitude of sEPSC in 1 h. (**h**) Summarized results of the amplitude of sEPSC from 5 different potentiation−induced protocols (black: baseline; red: 1 Hz for 60 s; blue: 1 Hz for 180 s; grey: 10 Hz for 10 s; green: 100 Hz for 1 s; orange: TBS).

**Figure 2 biomedicines-12-01072-f002:**
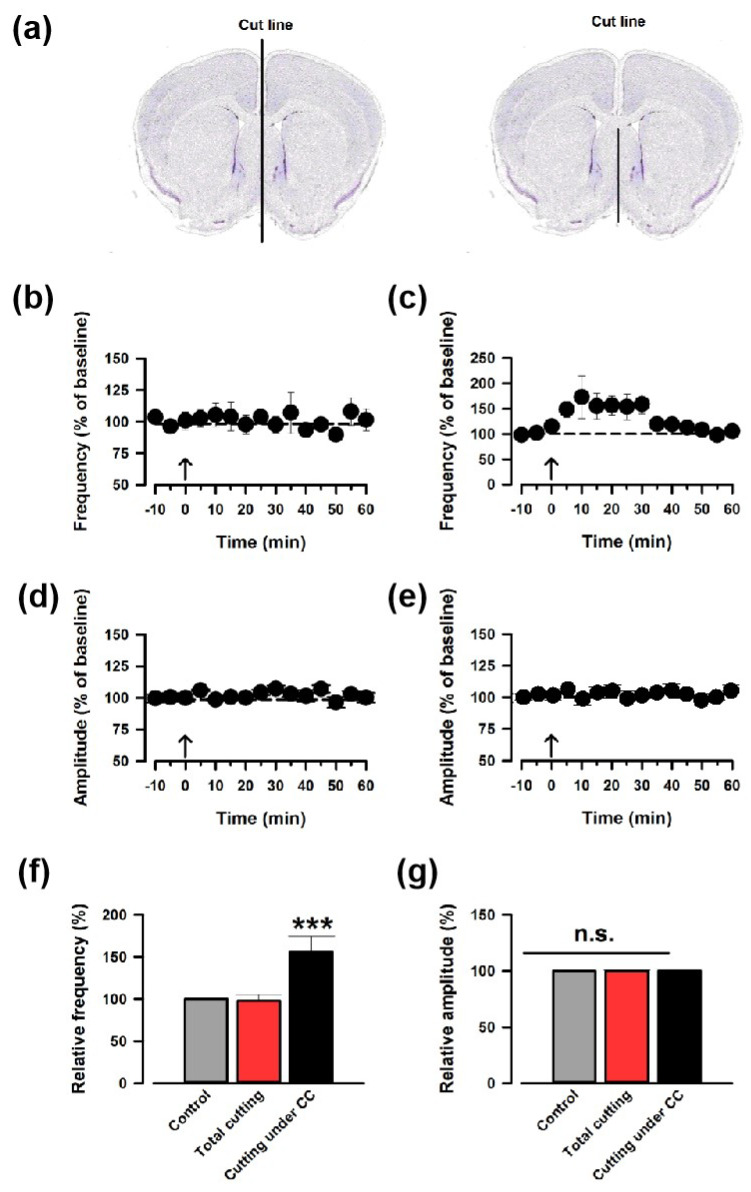
LFS-induced synaptic potentiation was induced through the CC pathway. (**a**) Two different ways to cut the coronal slices. The brain slices were cut through the midline (left) and cut under the CC (right). (**b**,**c**) The average frequency of sEPSC from slices that were cut through the midline (**b**) or cut under the CC. (**c**). The LFS was applied 10 min after the baseline recording (shown in arrow). (**d**,**e**) The averaged amplitude of sEPSC from slices that were cut through the midline (**d**) or cut under the CC (**e**). (**f**,**g**) Summarized results of the frequency of sEPSC from slices that were cut through the midline (**f**) or cut under the CC (**g**). *** *p* < 0.001, one-way ANOVA. n.s. not significant.

**Figure 3 biomedicines-12-01072-f003:**
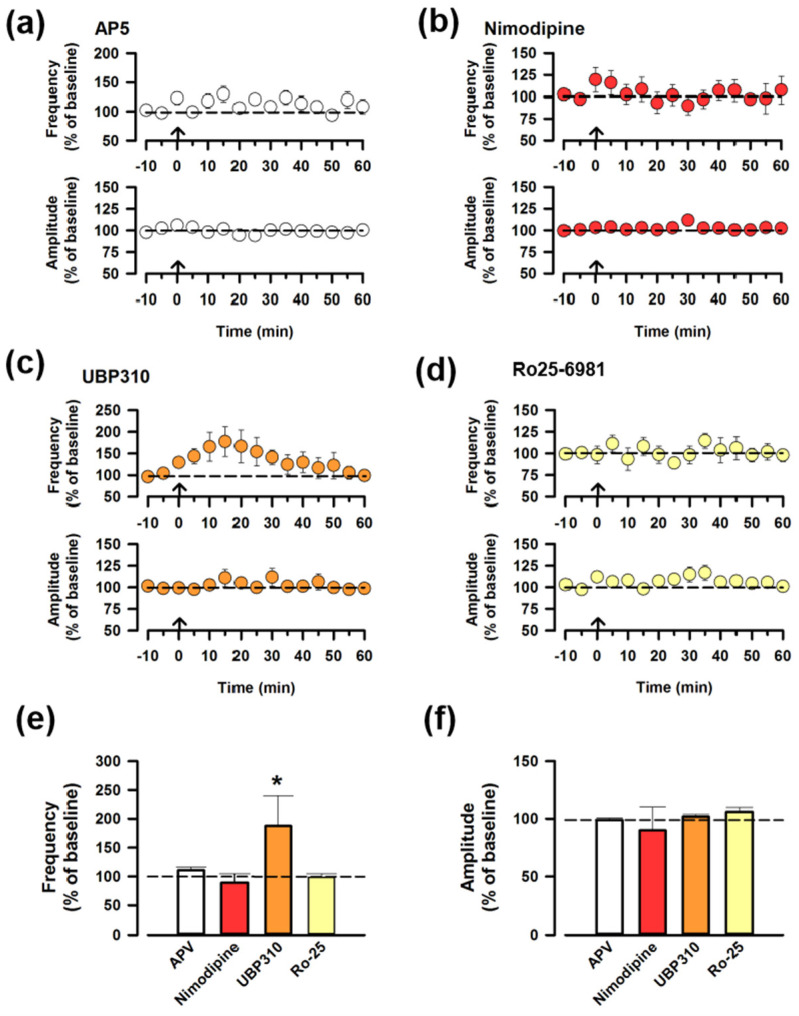
The synaptic potentiation of ACC-ACC is dependent on the NMDAR. (**a**) Averaged frequency and amplitude of sEPSC before and after the LFS applied through the ACC-ACC connection in the ACSF containing AP-5, the antagonist of NMDAR. The LFS was applied 10 min after the baseline recording (shown in arrow). (**b**) Averaged frequency and amplitude of sEPSC in the ACSF containing nimodipine, the blocker of the L-VDCC. (**c**) Averaged frequency and amplitude of sEPSC in the ACSF containing UBP310, the antagonist of the GluK1 receptor. (**d**) Averaged frequency and amplitude of sEPSC in the ACSF containing Ro25-6981, the antagonist of the GluN2B-containing NMDARs. (**e**) Summarized results of the frequency of sEPSC from different antagonists. * *p* < 0.05, one-way ANOVA. (**f**) Summarized results of the amplitude of sEPSC from different antagonists.

**Figure 4 biomedicines-12-01072-f004:**
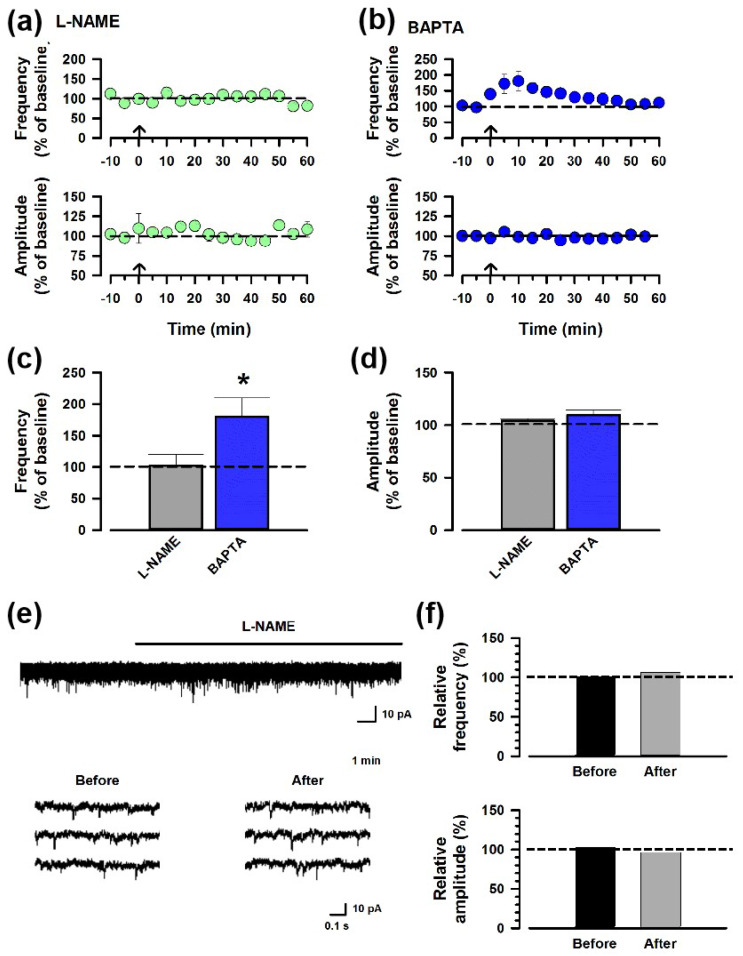
NO is required for the synaptic potentiation of the ACC-ACC connection. (**a**) Averaged frequency and amplitude of sEPSC before and after the LFS applied through the ACC-ACC connection in the ACSF containing L-NAME, the antagonist of NOS. The LFS was applied 10 min after the baseline recording (shown in arrow). (**b**) Averaged frequency and amplitude of sEPSC in the ACSF containing BAPTA, the Ca^2+^ chelator. (**c**,**d**) Summarized results of the frequency and amplitude of sEPSC applying L-NAME or BAPTA. * *p* < 0.05, *t*-test. (**e**) Sample traces of sEPSC showing L-NAME neither affected the frequency nor the amplitude of the sEPSC. (**f**) Summarized results of the frequency and amplitude of sEPSC before and after applying the L-NAME.

**Figure 5 biomedicines-12-01072-f005:**
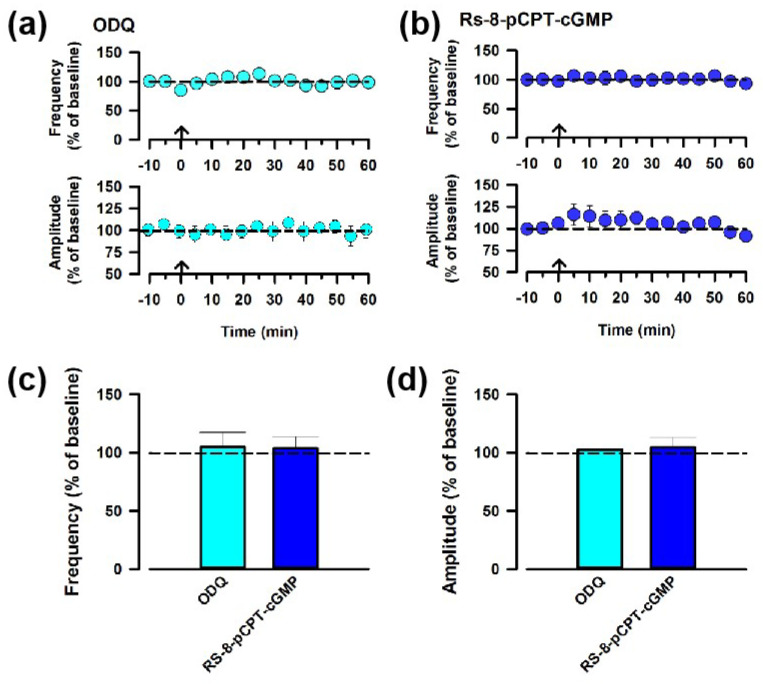
The sGC signaling pathway is required for NO-mediating ACC-ACC synaptic potentiation. (**a**) Averaged frequency and amplitude of sEPSC before and after the LFS applied through the ACC-ACC connection in the ACSF containing ODQ, the inhibitor of sGC. The LFS was applied 10 min after the baseline recording (shown in arrow). (**b**) Averaged frequency and amplitude of sEPSC in the ACSF containing Rp-8-pCPT-cGMP, the inhibitor of PKC. (**c**,**d**) Summarized results of the frequency and amplitude of sEPSC applying ODQ or Rp-8-pCPT-cGMP.

**Figure 6 biomedicines-12-01072-f006:**
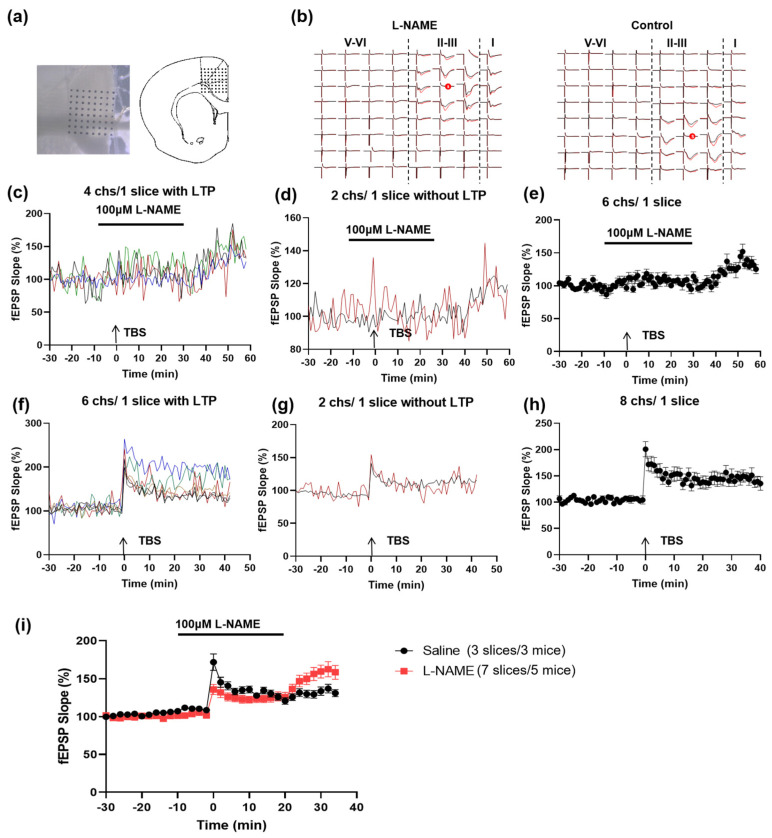
Local LTP in the ACC does not require NO. (**a**) Microphotograph and schematic diagram showing one example of ACC fEPSP recorded by using the MED64 system. A cortex slice containing the ACC was placed on a probe with 64 electrodes (MED-P515A, 8 × 8 array). One channel of the probe was selected as the stimulation site. The evoked field potentials in all the other channels were recorded 30 min before and 1 h after the TBS. (**b**) Two mapped figures show the evoked field potentials in the ACC with or without L-NAME perfusion. Field potentials were recorded from the other 63 channels 0.5 h before (black) and 3 h (red) after TBS was delivered to one channel marked ‘s’ with red circles. The fEPSP slope of 4 channels (differentiated by four different colors) with LTP (**c**) and 2 channels without potentiation (**d**) from 1 slice in L-NAME is shown, respectively. The average slope of the total 6 channels is shown in (**e**). The fEPSP slope of 6 channels with LTP (**f**) and 2 channels without potentiation (**g**) from 1 slice in the control condition is shown, respectively. The average slope of the total 8 channels is shown in (**h**). (**i**) Time course of the averaged fEPSP slope of all recorded channels in L-NAME and control ACSF (7 slices/5 mice in the L-NAME group and 3 slices/3 mice in the control group).

**Figure 7 biomedicines-12-01072-f007:**
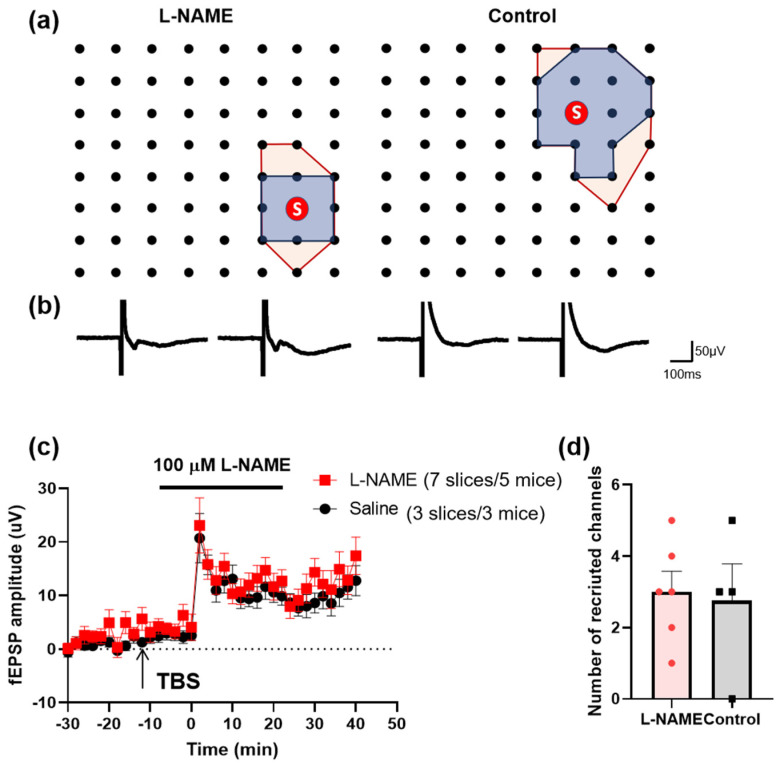
The recruitment of silent synapses after local LTP stimulation in the ACC does not require NO. (**a**) The polygonal diagram shows the baseline area of the activated channels with fEPSP (blue) and the enlarged area after TBS (red). The circled S indicates the stimulation site. This means that the recruited fEPSPs could be observed in the same channel in different slices (7 slices/5 mice in the L-NAME group and 3 slices/3 mice in the control group). (**b**) The superimposed traces indicate one channel showing recruited fEPSPs after TBS. (**c**) The amplitude of fEPSPs is summarized from all recruited channels (7 slices/5 mice in the L-NAME group, marked as red squares, and 3 slices/3 mice in the control group, marked as black circles). (**d**) The number of recruited channels is summarized after TBS induction.

## Data Availability

The raw data supporting the conclusions of this article will be made available by the authors on request.

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
