# Peer review of "The Pathway-Selective Dependence of Nitric Oxide for Long-Term Potentiation in the Anterior Cingulate Cortex of Adult Mice"

_biomedicines, 2024, doi:10.3390/biomedicines12051072_

Round 1

Reviewer 1 Report

Comments and Suggestions for Authors

In the manuscript, the authors reported an interesting study in which the authors demonstrated that the ACC-ACC potentiation could be selectively dependent on NO. They performed a number of experiments and that consistently suggested that the potentiation could be selective to the ACC-ACC pathway. The manuscript is well-prepared and the conclusions are supported with experimental results. Therefore, the manuscript could be recommended for publication in Biomedicines.

There are some minor issues found and that could be addressed further by the authors to improve further the overall quality of their work.

1. Citation. There are a quite number of conclusive descriptions in the manuscript. The authors do not cite properly references. For example, Line 56-57 and Lines 272-273.

2. The term for “LTP” is not consistent. Line 15 and Line 28.

3. Line 196: 10 microM. Space is needed.

4. Line 211-223: "Ca2+" is not presented consistently.

5. Discussion. Is there any reasons/factors that ACC-ACC potentiation is selectively dependent on NO? Could the authors please elaborate a bit further?

Author Response

Thank you very much for taking the time to review this manuscript. Please find the detailed responses below and the corresponding revisions in the re-submitted files.

  1. There are a quite number of conclusive descriptions in the manuscript. The authors do not cite properly references. For example, Line 56-57 and Lines 272-273.

Thank you for pointing this out. We have added citations to those sentences in the revised manuscript.

  1. The term for “LTP” is not consistent. Line 15 and Line 28.

We agree with this comment. We have corrected this mistake in the revised manuscript.

  1. Line 196: 10 microM. Space is needed.

Thank you for this suggestion. We have corrected this mistake in the revised manuscript.

  1. Line 211-223: "Ca2+" is not presented consistently.

Thank you for this suggestion. We have corrected this mistake in the revised manuscript.

  1. Discussion. Is there any reasons/factors that ACC-ACC potentiation is selectively dependent on NO? Could the authors please elaborate a bit further?

Thank you for this suggestion. In our study, we found that bath application of the selective inhibitor for the nitric oxide synthase (NOS) blocked the synaptic potentiation mediated by the ACC-ACC connection. As a comparison, we induced the LTP by local TBS stimulation in the ipsilateral ACC of the recorded neurons. Previous studies have reported that ACC neurons receive projections from other cortical areas as well as subcortical nuclei (Vogt BA, 2005; Bliss TV, 2016). Those corticocortical projections include the ACC-ACC callosal projections. When the TBS was applied on the ACC, a variety of projections to the recorded ACC neuron can be activated. However, the induction of local TBS-induced LTP is not affected by the application of the L-NAME. Therefore, we proposed that the ACC-ACC potentiation is selectively dependent on NO. Additional description has been added in the revised manuscript.

Reviewer 2 Report

Comments and Suggestions for Authors

In the hippocampus, there is the synaptic long-term plasticity (LTP) that requires retrograde messengers (e.g. nitric oxide (NO), carbon oxide (CO) and arachidonic acid). Presynaptic LTP in the anterior cingulate cortex (ACC) has been shown to contribute to chronic pain-related anxiety. But it is still unknown whether this retrograde messenger-mediating LTP exists in the ACC. In this manuscript, the authors investigate the role of NO in the synaptic potentiation in the pyramidal neurons in the ACC, especially in the ACC-ACC potentiation.

They showed that ONLY low-frequency stimulation (LFS, 1 Hz for 60 s) facilitates contralateral facilitation of the frequency instead of the amplitude of sEPSC (figure 1), and the ACC-ACC potentiation is mediated by the projections in the corpus callosum (CC) (figure 2), and the synaptic potentiation of ACC-ACC requires NMDAR, L-VDCC, GluN2B (figure 3), and NO (figure 4), and cGMP/PKG signaling (figure 5). But, BAPTA in the recording pipette FAILED to block the potentiation (figure 4b and c). Furthermore they showed that NO is NOT required for homo-LTP within the ACC using the multi-electrode recording system (figure 6 and 7).

Based on the above results they propose that LFS-induced ACC-ACC potentiation but not homo-LTP in the ACC is mediated by NO signals and that the effect of postsynaptic NO is derived from besides CC-producing neurons. Although the reasons why the postsynaptic inhibition of Ca2+ by BAPTA did NOT block the ACC-ACC potentiation in figure 4 b and c are obscured in this experiment, I think this work is well organized and makes a nice contribution to the understanding of the role of NO in the LTP in the ACC and in the ACC-ACC potentiation.

I have some concerns to address as below.

Major points

1.    They state this indicates that the presynaptic glutamate release was enhanced by the LFS (lane 144 and 155). They need to explain the reason why they state so.

2.    They would better off presenting the data of the control experiments of the synaptic potentiation of ACC-ACC connection without any inhibitors in the bath solution (figures 3-5).

3.    Is the NOS present in either pre- or post-synaptic neurons in ACC?

4.    What are the differentiated roles of the ACC-ACC potentiation and the LTP in the ACC in the coding of pain and unpleasantness?

Minor points

1.    (sEPSC (Fig. 1a) should be (sEPSC) in lane 138.

2.    Fig. 1F and 1H should be Fig. 1f and 1h in lanes 147 and 148.

3.    CPSC should be sEPSC in lane 156.

4.    “indicating that NO in involved in the synaptic potentiation of the ACC-ACC connection” should be “indicating that NO is involved in the synaptic potentiation of the ACC-ACC connection” (lanes 220 and 221).

5.    decreased PPR should be decreased paired-pulse ratio (lane 334).

Author Response

Thank you very much for taking the time to review this manuscript. Please find the detailed responses below:

  1. They state this indicates that the presynaptic glutamate release was enhanced by the LFS (lane 144 and 155). They need to explain the reason why they state so.

Thank you for this question. The interpretation of sEPSC amplitude is complicated by the fact that these events represent a mixture of miniature and evoked (by spontaneous cell firing) synaptic currents. Early studies have proved that the frequency of the sEPSC represent the presynaptic release of the neurotransmitters (Kevin J. Staley; Istvan Mody, 1991). It has been shown that, in hippocampal cultures, activity blockade by NBQX treatment led to shorter decay time, increased amplitude and frequency of the AMPA mEPSC (Thiagarajan et al., 2002; Thiagarajan et al., 2005). To find out whether this form of activity-dependent plasticity exists in the cortical cultures, We performed whole-cell patch-clamp recording here and patch the neurons at-70 mV, therefore, the increased frequency of the sEPSC indicates the enhanced probability of the spontaneous glutamate release.

  1. They would better off presenting the data of the control experiments of the synaptic potentiation of ACC-ACC connection without any inhibitors in the bath solution (figures 3-5).

Thank you for this suggestion. We have shown the data without any inhibitors in figure 1 and Figure2c,2e. The recording with inhibitors and control were performed alternatively. We just summarized the control data and put them as Figure 1.

  1. Is the NOS present in either pre- or post-synaptic neurons in ACC?

Thank you for this good question. It is major finding that NMDAR and neuronal NOS (nNOS) exist in the postsynaptic site. It was widely proposed the hypothesis which NO, as a retrograde messenger, diffuses from postsynaptic to presynaptic site. However , we cannot rule out the possibility that the NOS can also present in presynaptic neurons. In a recent study, by using the double patch-clamp techniques, the presynaptic LTP was blocked by either perfusing the L-NAME or injected MK-801 in the presynaptic neurons in the insular cortex, which indicates that presynaptic NMDARs as well as its following NOS activation may also exist.

  1. What are the differentiated roles of the ACC-ACC potentiation and the LTP in the ACC in the coding of pain and unpleasantness?

Thank you for this question. By using the animal model of neuropathic pain, two recent studies about the ACC-ACC connection have reported that the ACC-ACC connection modulate both the ipsilateral and contralateral hindpaw hypersensitivity of the nerve injured side as well as the pain related unpleasantness. There are several possible positive loops exist in the CNS to enhance or facilitate pain-related information: short-distance positive feedback (ACC-ACC); long distance positive feedback (ACC-spinal cord); unilateral positive feedback and bilateral positive feedback. Considering the circuit-like innervation between bilateral ACCs, such unique circuits provide a positive feedback loop for the processing of nociceptive information in the cortex. Thus, pain perception can be reinforced through such short-distance and long-distance projections. We can infer that the long-term hyperactivity is related to the enhanced synaptic potentiation between ACC-ACC connection. The role of LTP in the ACC in the coding of pain and unpleasantness have been reported in cumulative studies. LTP was recorded in the ACC vivo in the contralateral side of the digit amputation in anesthetized rats. In the in vitro ACC slices, LTP was occluded.

Minor points

  1. (sEPSC (Fig. 1a) should be (sEPSC) in lane 138.

  1. Fig. 1F and 1H should be Fig. 1f and 1h in lanes 147 and 148.

  1. CPSC should be sEPSC in lane 156.

  1. “indicating that NO in involved in the synaptic potentiation of the ACC-ACC connection” should be “indicating that NO is involved in the synaptic potentiation of the ACC-ACC connection” (lanes 220 and 221).

  1. decreased PPR should be decreased paired-pulse ratio (lane 334).

Thank you for these suggestions. We have corrected these mistakes in the revised manuscript.

Reviewer 3 Report

Comments and Suggestions for Authors

Submitted manuscript is a nicely planned and executed research paper. The authors used  in vitro approach, including patch-clamp technique, to assess the effect of NO in the synaptic potentiation in the pyramidal neurons of the anterior cingulate cortex.

I have some minor comments:

1) The impact of NO on spontaneous excitatory synaptic transmission, was studied with the use of NOS inhibitor, L-NAME (100 μM) what resulted in
neither the frequency nor the amplitude of sEPSCs being changed.

However, we cannot exclude that the application of NO donor would yield a change in sEPSC - please, comment on that.

2) The language requires editing

Comments on the Quality of English Language

Manuscript requires editing, below some examples

line 30 - At least there are three different....

lines 39-40 By pairing the electric stimulation on the presynaptic fibers with the NO/CO application, LTP was recorded in the CA1, this LTP is not dependent to the NMDARs and L-VDCCs ???

 47 Early studies using inhibitors of NOS extracellularly or intracellularly in
the hippocampus found that the LTP was blocked

76 picrotocin - I presume, picrotoxin

86 The whole brain tissue was cooled for

Author Response

Thank you very much for taking the time to review this manuscript. Please find the detailed responses below

1) The impact of NO on spontaneous excitatory synaptic transmission, was studied with the use of NOS inhibitor, L-NAME (100 μM) what resulted in

neither the frequency nor the amplitude of sEPSCs being changed.

However, we cannot exclude that the application of NO donor would yield a change in sEPSC - please, comment on that.

It has been reported that the application of NO donner enhanced mEPSCs in hippocampal neurons as well as the insular cortex. We did not rule out the possibility that the NO donner would yield a change in the sEPSC of the ACC. This requires to be confirmed in the future study.

2) The language requires editing

Comments on the Quality of English Language

Manuscript requires editing, below some examples

line 30 - At least there are three different....

lines 39-40 By pairing the electric stimulation on the presynaptic fibers with the NO/CO application, LTP was recorded in the CA1, this LTP is not dependent to the NMDARs and L-VDCCs ???

 47 Early studies using inhibitors of NOS extracellularly or intracellularly in

the hippocampus found that the LTP was blocked

76 picrotocin - I presume, picrotoxin

86 The whole brain tissue was cooled for

Thank you for these comments. We have corrected these mistakes and improved the English editing in the revised manuscript.

Round 2

Reviewer 2 Report

Comments and Suggestions for Authors

In my opinion, in this revised manuscript, the responses to the major points raised previously were adequate and now this is acceptable for publication in Biomedicines.